# The Influence of Methylating Mutations on Acute Myeloid Leukemia: Preliminary Analysis on 56 Patients

**DOI:** 10.3390/diagnostics10050263

**Published:** 2020-04-29

**Authors:** Sergiu Pasca, Cristina Turcas, Ancuta Jurj, Patric Teodorescu, Sabina Iluta, Ionut Hotea, Anca Bojan, Cristina Selicean, Bogdan Fetica, Bobe Petrushev, Vlad Moisoiu, Alina-Andreea Zimta, Valentina Sas, Catalin Constantinescu, Mihnea Zdrenghea, Delia Dima, Ciprian Tomuleasa

**Affiliations:** 1Department of Hematology, Iuliu Hatieganu University of Medicine and Pharmacy, 400124 Cluj Napoca, Romania; 2Research Center for Functional Genomics and Translational Medicine, Iuliu Hatieganu University of Medicine and Pharmacy, 400124 Cluj Napoca, Romania; 3Department of Hematology, Ion Chiricuta Clinical Cancer Center, 400006 Cluj Napoca, Romania; 4Medfuture Research Center for Advanced Medicine, Iuliu Hatieganu University of Medicine and Pharmacy, 400124 Cluj Napoca, Romania

**Keywords:** acute myeloid leukemia, methylation, classification, TCGA, mutations

## Abstract

Acute myeloid leukemia (AML) is a hematologic malignancy characterized by abnormal proliferation and a lack of differentiation of myeloid blasts. Considering the dismal prognosis this disease presents, several efforts have been made to better classify it and offer a tailored treatment to each subtype. This has been formally done by the World Health Organization (WHO) with the AML classification schemes from 2008 and 2016. Nonetheless, there are still mutations that are not currently included in the WHO AML classification, as in the case of some mutations that influence methylation. In this regard, the present study aimed to determine if some of the mutations that influence DNA methylation can be clustered together regarding methylation, expression, and clinical profile. Data from the TCGA LAML cohort were downloaded via cBioPortal. The analysis was performed using R 3.5.2, and the necessary packages for classical statistics, dimensionality reduction, and machine learning. We included only patients that presented mutations in *DNMT3A, TET2, IDH1/2, ASXL1, WT1, and KMT2A*. Afterwards, mutations that were present in too few patients were removed from the analysis, thus including a total of 57 AML patients. We observed that regarding expression, methylation, and clinical profile, patients with mutated *TET2, IDH1/2,* and *WT1* presented a high degree of similarity, indicating the equivalence that these mutations present between themselves. Nonetheless, we did not observe this similarity between *DNMT3A-* and *KMT2A*-mutated AML. Moreover, when comparing the hypermethylating group with the hypomethylating one, we also observed important differences regarding expression, methylation, and clinical profile. In the current manuscript we offer additional arguments for the similarity of the studied hypermethylating mutations and suggest that those should be clustered together in further classifications. The hypermethylating and hypomethylating groups formed above were shown to be different from each other considering overall survival, methylation profile, expression profile, and clinical characteristics. In this manuscript, we present additional arguments for the similarity of the effect generated by *TET2*, *IDH1/2,* and *WT1* mutations in AML patients. Thus, we hypothesize that hypermethylating mutations skew the AML cells to a similar phenotype with a possible sensitivity to hypermethylating agents.

## 1. Introduction

Acute myeloid leukemia (AML) is a hematologic malignancy characterized by aberrant differentiation and abnormal proliferation of myeloid blasts [1,2,3]. Continuous efforts are being made to uncover the biology of AML, with implications both in prognosis and in tailoring new therapeutic options for these patients. These efforts can be formally observed in the AML World Health Organization (WHO) classification in 2008 and 2016 [4,5].

Over time, several research groups have shown different prognostic capabilities of expression and methylation profiling [6,7,8]. Nonetheless, most molecular biology markers used in clinical practice are represented by gene mutations, while DNA methylation and RNA expression are commonly only used within the research setting [6,9,10,11,12]. This might be due to economic reasons, levels of evidence, and the reproducibility that can be achieved. Nevertheless, the epigenome and transcriptome are known to be different between AML patients, having a good potential to act as tools for classifying AML and choosing the best available therapy. Thus, an inferred epigenome and transcriptome considering mutations that affect genes implicated in methylation could be useful to improve our understanding of the biology of AML as well as the AML clinical practice.

Important pathways in AML are formed by key genes implicated in methylation, hydroxymethylation, and demethylation [13,14,15,16]. These are represented by, but not limited to, *DNMT3A, TET2, IDH1/2, WT1, ASXL1,* and *KMT2A*. *DNMT3A* is a de novo DNA methyl transferase that in physiological conditions transforms a non-methylated CpG into a hemymethylated CpG. In AML, it has been shown that *DNMT3A* can present mutations in an important portion of cases, which induce a global hypomethylation, changes in AML biology, and a worse prognosis [17]. *TET2* physiologically leads to the hydroxymethylation of methylated CpG, initiating the demethylation process. Most mutations in *TET2* lead to the formation of non-functional forms of *TET2*, associated with a decrease in its function and a hypermethylated genome [15]. In contrast to *TET2, IDH1/2* present few mutations that can be commonly observed in AML and that induce the formation of 2-hydroxy-glutarate, a metabolite that inhibits *TET2*, leading to similar effects to *TET2* mutations [15]. *WT1* is a gene that physically interacts with *TET2,* aiding in the process of hydroxymethylation. As in the case of *TET2*, most mutations observed in *WT1* lead to the formation of a non-functional protein, indirectly determining a diminished activity of *TET2* [18]. *ASXL1* is a gene that has been observed to be important in interacting with TET2 and can be mutated in several myeloid malignancies [19]. *KMT2A* presents roles in methylating H3K4, DNA binding, and either inhibiting methylation or modifying the chromatin conformation. Mutations in this genes are generally represented either by amplifications or fusions, with patients presenting a hypomethylated epigenome and generally having a worse prognosis [20,21,22,23].

The aim of this study was to determine the differences that the presented mutations induce in AML regarding methylation, expression, and clinical profile.

## 2. Material and Methods

### TCGA Analysis

cBioPortal was used to download clinical, methylation, and expression data from the TCGA cohort on adult AML [24,25,26]. The results shown here are based upon data generated by the TCGA Research Network: https://www.cancer.gov/tcga (accessed on 21 January 2020). Data analysis was performed using R version 3.5.2. Oncoprints were generated using the cBioPortal platform. The normality of the data was assessed using the Shapiro-Wilk test and histogram visualization. For the comparison of two groups with continuous variables, we used the Mann-Whitney-Wilcoxon test. For comparing multiple groups with continuous variables, we used the Kruskal-Wallis test. The analysis of contingency tables was performed using Fisher’s test. Survival analysis was performed using Kaplan-Meier curves and the log-rank test. Unsupervised machine learning was performed using pheatmap and prcomp to generate the needed heatmaps and principal component analysis (PCAs), respectively. Supervised machine learning was performed using randomForest. The differential expression analysis was performed using the DEseq2 package [27], with an lfcThreshold of 0.32 and using the Benjamini-Hochberg p adjustment method. Functional enrichment analysis was performed using FENet [28], topGO, and GO.db [29]. A *p* value under 0.05 was considered statistically significant.

## 3. Results

### 3.1. Patient Selection

We included the patients that had transcriptomic and methylation profiles available and presented mutations in any of the following genes: *TET2, IDH1, IDH2, WT1, KMT2A, DNMT3A, ASXL1*. After applying these criteria, 85 patients were initially included (Figure 1A). Because these genes are known to highly influence DNA methylation and thus expression [13,15,18,30,31,32], we further included only the patients in which only one of the seven genes was mutated, so that the influence of each mutation could be observed without the influence of the other mutations included. After applying this second filter, the resulting cohort included 57 patients. Considering that *ASXL1* was mutated only in one patient, that patient was excluded from the analysis to avoid overfitting, thus using a cohort of 56 AML patients for further analyses (Figure 1B).

### 3.2. Clinical Data and Survival Analysis

The available clinical parameters analyzed were represented by gender, age, French-American-British (FAB) classification transposed in the equivalent WHO-NOS (Not Otherwise Specified), white blood cell count (WBC), bone marrow blast percentage, peripheral blood blast percentage, and cytogenetic risk. Clinically, patients with hypermethylating mutations had a higher peripheral blast percentage and included more AML patients without maturation compared with the hypomethylating mutations group, the latter including acute monoblastic/monocytic leukemia more frequently. The cytogenetic risk most frequently reported in both hypermethylating (83%) and hypomethylating (84%) groups was represented by intermediate cytogenetic risk (Table 1).

In subgroup analysis, in the hypermethylating group, the only difference between the four mutations was in age (Table 2), whereas in the hypomethylating group, there were no differences in the clinical parameters (Table 3).

Regarding survival analysis, the hypomethylating group showed a lower overall survival (OS) (*p* = 0.007), but there was no difference in disease-free survival (DFS). In the hypermethylating group, there was no significant difference in OS between mutations, but there was a statistically significant difference in DFS (*p* = 0.018) between mutations. Analyzing the Kaplan-Meyer curves, it appears that this difference was caused mainly by *WT1* mutated patients (*n* = 5), while the other mutations showed a similar trend between each other. In the hypomethylating group, there was no difference in survival between mutations (Figure 2).

### 3.3. Clustering and Principal Component Analysis

For the methylation profile, a certain enrichment of hypermethylating and hypomethylating mutations, respectively, was observed between the two most distant clusters on the heatmap. This discrimination between the hypermethylating and hypomethylating groups is lower when considering the expression profile.

In Figure 3 we depicted the heatmaps and PCAs for the methylation and expression profiles.

### 3.4. Supervised Machine Learning

When analyzing the predictive capabilities between hypermethylating and hypomethylating groups, hypermethylating mutations were determined with an accuracy of 75%, and hypomethylating mutations were determined with an accuracy of 81.25% with an out of bag (OOB) of 21.43%.

When only hypermethylating mutations were introduced in the algorithm, the accuracy varied from 0 to 14.3%, and the OOB was 91.65%. When only hypomethylating mutations were introduced in the algorithm, all mutations were classified as *KMT2A* with an OOB of 71.88%.

### 3.5. Differential Expression Analysis and Functional Enrichment Analysis

When comparing the hypermethylating with the hypomethylating group without setting a threshold, 484 genes were upregulated and 647 were downregulated. When the lfcThreshold was set, 89 genes were upregulated and 103 were downregulated. When the latter genes were analyzed with topGO, 213 processes were found to be different between the two groups. When comparing *WT1* with *TET2* + *IDH1/2,* we observed that without setting a threshold 15 genes were upregulated and 74 were downregulated. When the lfcThreshold was set, 3 genes were upregulated and 31 were downregulated. When the latter genes were analyzed with topGO, 43 processes were found to be different between the two groups.

When comparing *TET2* with *IDH1/2,* we observed that without setting a threshold 74, genes were upregulated and 31 were downregulated. When the lfcThreshold was set, 34 genes were upregulated and 10 were downregulated. When the latter genes were analyzed with topGO, 40 processes were found to be different between the two groups. When comparing *IDH1* with *IDH2* without a threshold set, 5 genes were upregulated and 14 were downregulated. When the lfcThreshold was set, there were no genes upregulated and 5 were downregulated. When the latter genes were analyzed with topGO, 92 processes were found to be different between the two groups. The counterintuitive number of processes was caused by three differentially expressed *HLA-D* family genes. When comparing *DNMT3A* with *KMT2A* without a threshold, 90 genes were upregulated and 84 were downregulated. After setting the lfcThreshold, 40 genes were upregulated and 35 were downregulated. When the latter genes were analyzed with topGO, 157 processes were found to be different between the two groups.

On the functional networks generated, the network with most nodes and edges was generated by the DE genes between the hypermethylating and hypomethylating groups, followed by the comparison between *DNMT3A* and *KMT2A*. After them, the networks formed in the hypermethylating group had a clear drop in nodes and edges, giving another argument for the similarity between the genes selected for the hypermethylating group (Figure 4).

## 4. Discussion

In the current study, we have shown that mutations in *TET2, IDH1/2,* and *WT1* present similar methylation, expression, and clinical profiles, making a case for these mutations to be clustered together in further AML classifications.

Both in the case of *IDH1* R132 and of *IDH2* R140 or R172, it has been shown that they switch *IDH1/2* from generating α-keto-glutarate into generating D-2-hydroxyglutarate [15,33,34]. The generated metabolite, 2-hydroxiglutarate, has been shown to have an inhibitory effect on *TET2*, leading to similar effects to the loss of function that *TET2* can present [35,36]. Other studies on *IDH1/2* mutations have shown that these can induce an impaired differentiation in AML blasts, thus explaining the more frequent immature forms of AML that occur in the hypermethylating group [15,35,37].

*WT1* mutations are generally represented by loss-of-function mutations [38]. It has been shown that truncated *WT1* messenger RNA is degraded, thus reducing the amount of protein produced [39]. Because of the known interaction between WT1 and TET2, a reduction in WT1 would potentially reduce TET2 activity, a fact that could be observed indirectly in the current study [18]. 

In accordance with published data, we consider that a large amount of evidence shows the equivalence between *TET2, IDH1/2,* and *WT1* mutations in AML, with similar global and gene-specific CpG methylation profiles between them and similar expression profiles.

Wild-type *DNMT3A* acts as a de novo DNA methyltransferase, and its mutations are generally associated with hypomethylation compared with the wild-type allele, especially in the case of R882 mutations [30]. Because DNMT3A mutations generally lead to an impaired function of DNMT3A leading to globally hypomethylated genome, it can be hypothesized that AML presenting DNMT3A mutations will have a different methylation and expression profile compared with the hypermethylating group. This hypothesis was confirmed in our study, as the hypomethylating and hypermethylating groups present different WHO-NOS subtypes, OS, CpG methylation, and expression profiles. An association of *DNMT3A* mutations with myelomonocytic or monoblastic/monocytic AML was previously observed and could potentially be used as an argument of the validity of these results [40].

*KMT2A* has different modes of action, the most prominent known so far being its role in methylating H3K4 [41]. Nevertheless, it has also been shown that KMT2A binds to unmethylated DNA and inhibits methylation [23]. Thus, the most common mutations observed in these gene, amplifications, should act as an inhibitor of CpG methylation [20,22,23,41,42,43]. Considering the results of this study, it cannot be suggested that mutations in KMT2A have a similar effect to those in DNMT3A, as the expression profile presents marked differences between the two gene mutations.

The presented mutations are mostly associated with intermediate cytogenetic risk [44], both in the literature and in our study. Thus, this classification can be better tailored in the future for patients with intermediate cytogenetics AML [17].

Because of the role that the included mutations have in modifying CpG methylation, it could be hypothesized that AML presenting different mutations will respond different to hypomethylating agents (HMA). The HMAs currently used in clinical practice outside clinical trials are azacytidine (AZA) and decitabine (DAC).

The AZA mechanism of action is based on DNA hypomethylation. Of the administered AZA, 10–20% is converted to 5-aza-2′-deoxycytidine triphosphate and then incorporated in the DNA strands [45]. Through the incorporation in DNA, it forms adducts between DNA and *DNMT1*. At high doses, the DNA strand is not able to recover, and apoptosis occurs, while at lower doses the adducts are degraded by the proteasome, and DNA synthesis is resumed in the absence of *DNMT1*, leading to hypomethylation [46]. The other approved hypomethylating agent is DAC, which has the same mechanism of action as AZA, but a part of the cellular metabolization is different when compared to AZA. Both compounds are considered equivalent and are referred to as HMAs.

For AML, HMA monotherapy was first used in patients over 65 years with palliative intent, with subsequent combinations being used in younger age groups [47]. Because of the mechanism of action, it was assumed that single-agent treatment would be heavily influenced by mutations that modify the DNA methylation profile, with *TET2* receiving more attention compared to other genes involved in DNA methylation, but the studies did not reach a consensus. Currently, the clinical prognosis of patients treated with HMAs does not take into account *TET2* or other equivalent mutations such as *IDH1/2* or *WT1* [48,49].

One of the more successful combinations for AML was represented by HMA plus venetoclax (VEN), the latter being a *BCL2* inhibitor first used for relapse/refractory (R/R) chronic lymphocytic leukemia (CLL). Because of the impressive results that VEN generated for CLL, it was further tested for other hematologic malignancies, of which AML presented unexpectedly outstanding results [50,51]. It has been shown that HMA plus VEN increased the overall response rate (ORR) to 76% in R/R AML, and in subgroup analysis of *IDH1/2* mutated AML, the ORR reached 82%, showing the importance of these mutations and their effect regarding the response to the HMA/VEN combination. In this case, the increase in ORR was mainly considered to be caused by the VEN component of the combination. Nonetheless, it should be considered whether the *IDH1/2* mutations also influence the response to HMAs in this instance.

Even if the current study presented *TET2, IDH1/2,* and *WT1* mutations as equivalent from some perspectives, it must be remembered that in some cases, these mutations might respond differently. One of the most notable examples of this is represented by the case of specific *IDH1/2* inhibitors such as ivosidenib and enasidenib. Ivosidenib (IVO) is an inhibitor of *IDH1* R132 that acts in synergy with AZA inducing myeloid blast differentiation [52,53], with promising results of this combination in the clinical scenario for relapsed/refractory (R/R) AML [54]. This synergy was also observed for the combination of HMA with the *IDH2* inhibitor enasidenib (ENA), both decreasing the level of DNA methylation in vitro and promoting blast differentiation [53,55,56].

The major limitation of the current study was the small patient cohort. The reason for not including other databases was the fact that most other datasets on AML do not present can; thus, we would not be able to assess *KMT2A* amplifications. Nevertheless, the results shown in our study are in agreement with other publications, especially in the case of hypermethylating mutations.

## 5. Conclusions

The hypermethylation and hypomethylation groups described above were shown to be different from each other considering OS, methylation profile, expression profile, and clinical presentation. In this manuscript, we brought additional arguments for the similarity of the effect generated by *TET2*, *IDH1/2,* and *WT1* mutations in AML patients. Thus, we hypothesized that hypermethylating mutations skew the AML cells to a similar phenotype with a possible sensitivity to hypermethylating agents, which should be further assessed clinically.

## Figures and Tables

**Figure 1 diagnostics-10-00263-f001:**
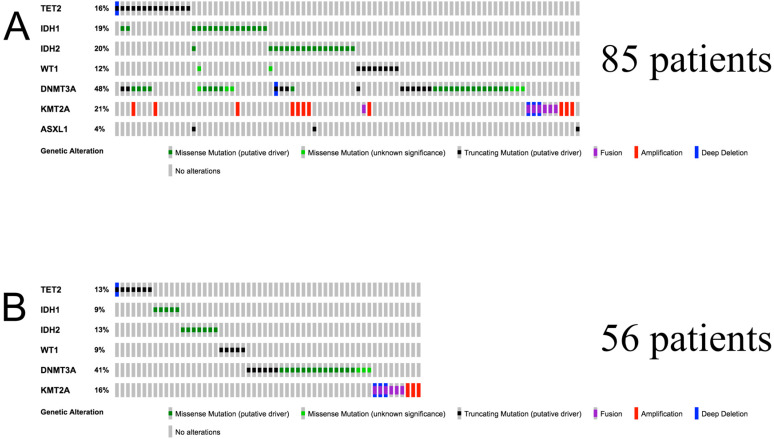
(**A**) Oncoprint representing patients before the mutual exclusivity condition was applied; (**B**) Oncoprint representing patients after the mutual exclusivity condition was applied.

**Figure 2 diagnostics-10-00263-f002:**
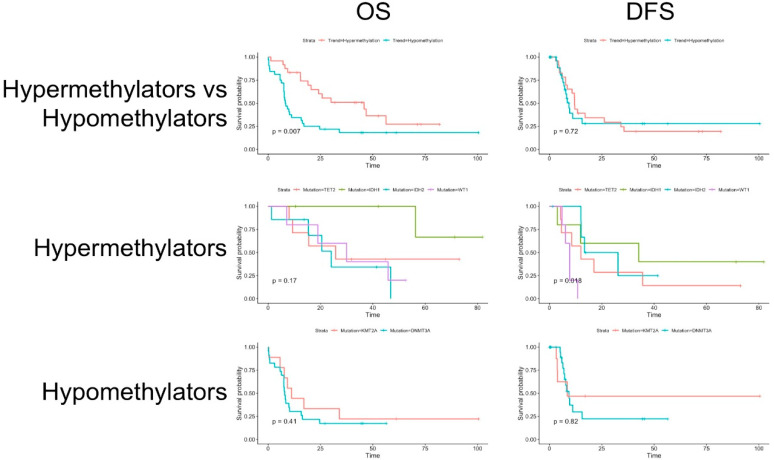
Kaplan-Meier curves showing the differences in overall survival (OS) and disease-free survival (DFS) between the hypermethylating and hypomethylating groups and in subgroup analysis between mutations.

**Figure 3 diagnostics-10-00263-f003:**
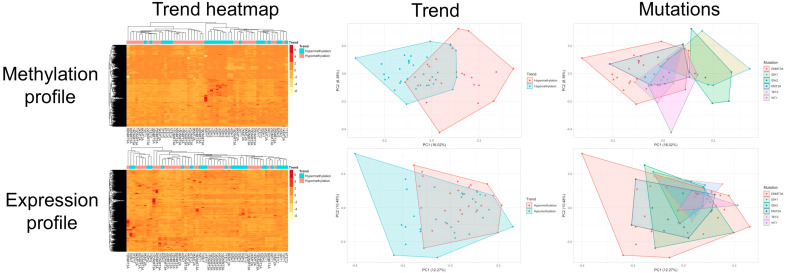
Heatmap and principal component analysis (PCA) representation of methylation and expression profiles. For methylation profile, the gene methylation score was used, while for expression profile, the transcript levels were used. The dendrograms show potential clustering (horizontal: between cases; vertical: between genes).

**Figure 4 diagnostics-10-00263-f004:**
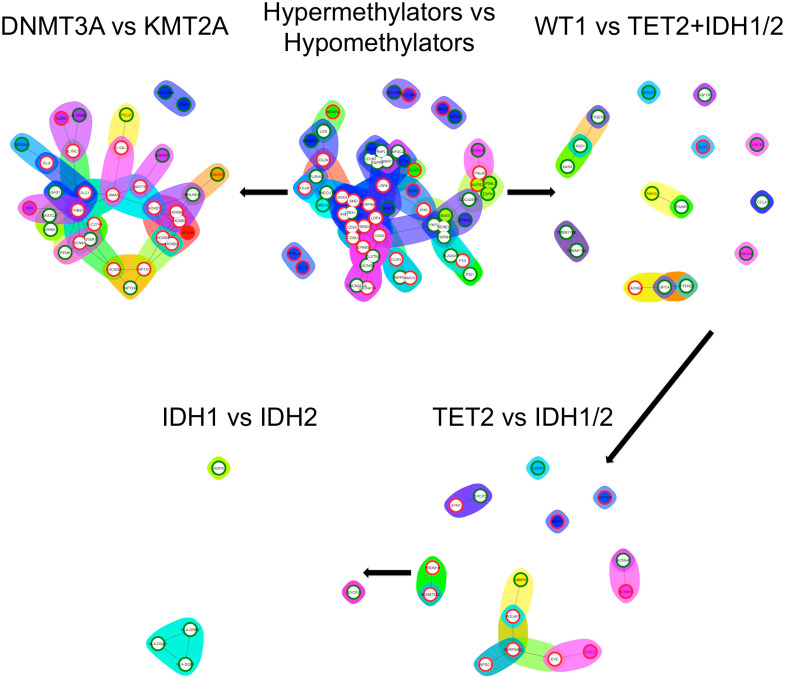
Functional network representation of the differentially expressed genes between groups. The second group from the titles was always considered as the baseline. Upregulation was represented with red circles, while downregulation was represented with green circles.

**Table 1 diagnostics-10-00263-t001:** Clinical differences between hypermethylators and hypomethylators (bold numbers are statistically significant data).

		Hypermethylating	Hypomethylating	*p* Value
		(*n* = 24)	(*n* = 32)	
Sex	Female	14 (58%)	15 (47%)	0.430
	Male	10 (42%)	17 (53%)	
Median age (quartile 1, quartile 3)		61 (38, 67)	58 (48, 66)	0.684
WHO NOS	AML with minimal maturation	3 (13%)	2 (6%)	**0.00271**
	AML without maturation	10 (42%)	4 (13%)	
	AML with maturation	6 (25%)	5 (16%)	
	Acute myelomonocytic leukemia	5 (21%)	9 (28%)	
	Acute monoblastic/monocytic leukemia	0 (0%)	11 (34%)	
	Acute megakaryocytoblastic leukemia	0 (0%)	1 (3%)	
Median WBC (quartile 1, quartile 3) (/μL)		16.95 (5.40, 60.33)	42.70 (8.13, 91.20)	0.224
Median bone marrow blast percentage (quartile 1, quartile 3)		76 (59, 87)	76 (57, 86)	0.734
Median peripheral blood blast percentage (quartile 1, quartile 3)		51 (17, 86)	10 (4, 58)	**0.0048**
Cytogenetic risk	Good	2 (8%)	0 (0%)	0.388
	Intermediate	20 (83%)	27 (84%)	
	Poor	1 (4%)	4 (13%)	
	Not determined	1 (4%)	1 (3%)	

**Table 2 diagnostics-10-00263-t002:** Clinical differences between different hypermethylating mutations (bold numbers are statistically significant data).

		TET2	IDH1	IDH2	WT1	*p* Value
		(*n* = 7)	(*n* = 5)	(*n* = 7)	(*n* = 5)	
Gender	Female	5 (71%)	2 (40%)	5 (71%)	2 (40%)	0.525
	Male	2 (29%)	3 (60%)	2 (29%)	3 (60%)	
Median age (quartile 1, quartile 3)		61 (49, 72)	32 (27, 38)	67 (62, 70)	57 (53, 61)	**0.0118**
WHO NOS	AML with minimal maturation	0 (0%)	0 (0%)	2 (29%)	1 (20%)	0.153
	AML without maturation	3 (43%)	5 (100%)	1 (14%)	1 (20%)	
	AML with maturation	3 (43%)	0 (0%)	2 (29%)	1 (20%)	
	Acute myelomonocytic leukemia	1 (14%)	0 (0%)	2 (29%)	2 (40%)	
	Acute monoblastic/monocytic leukemia	0 (0%)	0 (0%)	0 (0%)	0 (0%)	
	Acute megakaryoblastic leukemia	0 (0%)	0 (0%)	0 (0%)	0 (0%)	
Median WBC (quartile 1, quartile 3) (/μL)		9.80 (3.95, 30.75)	39.80 (8.20, 63.70)	11.50 (3.80, 38.40)	27.70 (27.10, 61.60)	0.339
Median bone marrow blast percentage (quartile 1, quartile 3)		63 (51, 86)	86 (85, 91)	72 (57, 81)	72 (61, 86)	0.255
Median peripheral blood blast percentage (quartile 1, quartile 3)		32 (15, 60)	85 (83, 88)	43 (14, 68)	52 (49, 63)	0.164
Cytogenetic risk	Good	1 (14%)	0 (0%)	0 (0%)	1 (20%)	0.7187
	Intermediate	6 (84%)	4 (80%)	6 (84%)	4 (80%)	
	Poor	0 (0%)	0 (0%)	1 (14%)	0 (0%)	
	Not determined	0 (0%)	1 (20%)	0 (0%)	0 (0%)	

**Table 3 diagnostics-10-00263-t003:** Clinical differences between the hypomethylating mutations.

		DNMT3A	KMT2A	*p* Value
		(*n* = 23)	(*n* = 9)	
Gender	Female	13 (57%)	2 (22%)	0.122
	Male	10 (43%)	7 (78%)	
Median age (quartile 1, quartile 3)		58 (50, 71)	54 (45, 64)	0.571
WHO NOS	AML with minimal maturation	0 (0%)	2 (22%)	
	AML without maturation	4 (17%)	0 (0%)	
	AML with maturation	4 (17%)	1 (11%)	
	Acute myelomonocytic leukemia	6 (26%)	3 (33%)	
	Acute monoblastic/monocytic leukemia	8 (35%)	3 (33%)	
	Acute megakaryoblastic leukemia	1 (4%)	0 (0%)	
Median WBC (quartile 1, quartile 3) (/μL)		75.20 (15.15, 98.70)	8.40 (2.30, 25.90)	0.00497
Median bone marrow blast percentage (quartile 1, quartile 3)		76 (55, 86)	75 (67, 83)	0.949
Median peripheral blood blast percentage (quartile 1, quartile 3)		11 (6, 76)	0 (0, 14)	0.0477
Cytogenetic risk	Good	0 (0%)	0 (0%)	0.0572
	Intermediate	21 (91%)	6 (66%)	
	Poor	1 (4%)	3 (33%)	
	Not determined	1 (4%)	0 (0%)	

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
