# Peer review of "The Influence of Methylating Mutations on Acute Myeloid Leukemia: Preliminary Analysis on 56 Patients"

_diagnostics, 2020, doi:10.3390/diagnostics10050263_

Round 1
Reviewer 1 Report
The paper by Pasca et al. report the analysis of 57 AML patients to assess the role of methylating mutations in patients’ prognostic classification and therapy.
The topic is interesting however, the manuscript’s structure lacks of coherence and it is difficult to follow.
In addition, grammar structures and typo errors should be carefully reviewed throughout the paper.
Specific Comments:
Abstract: abstract structure is unclear and difficult to follow and unclear. It should be restructured and largely rewritten. See in particular lines 18-23; lines 27-29. Abbreviations should be specified at their first mention.
Line 23: were
Lines 46-49: these questions will be assessed? They are the aims of the work; thus, the results and discussion should describe the answers.
Line 50: as are…. this sentence is not clear. Please rephrase
Lines 58-68: This section has to be completely rewritten. This is the first time that the AML genes are mentioned: their features and clinical associations should be better explained.
Lines 69-71: this section has not a conclusion.
Lines 90-92: Results description is unclear. What are the mutations found in in each patient? Are these mutations the same? Are these mutations already known?
Lines 103-104: delete this sentence, since it has been already stated.
Tables 2 and 3 “AML without maturation” should be listed before “AML with minimal maturation”
Figure legends should be improved to be more self-explicative.
Discussion should be restructured to answer to previously raised issue. More perspectives on clinical management and therapies should be also discussed.
Author Response
Reviewer 1
Comments and Suggestions for Authors
The paper by Pasca et al. report the analysis of 57 AML patients to assess the role of methylating mutations in patients’ prognostic classification and therapy.
The topic is interesting however, the manuscript’s structure lacks of coherence and it is difficult to follow.
In addition, grammar structures and typo errors should be carefully reviewed throughout the paper.
Thank you for the feed-back. We modified the text of the manuscript.
Specific Comments:
Abstract: abstract structure is unclear and difficult to follow and unclear. It should be restructured and largely rewritten. See in particular lines 18-23; lines 27-29. Abbreviations should be specified at their first mention.
Thank you for the feed-back. We restructured the abstract.
Line 23: were
Thank you for the feed-back. We changed “was” for “were” at line 23.
Lines 46-49: these questions will be assessed? They are the aims of the work; thus, the results and discussion should describe the answers.
Thank you for the feed-back. These questions do not represent the scope of the study and were intended as rhetorical questions to make the reader reflect to them. Nonetheless, we removed these lines from the current manuscript.
Line 50: as are…. this sentence is not clear. Please rephrase
Thank you for the feed-back. We modified the sentence so it would be more clear.
Lines 58-68: This section has to be completely rewritten. This is the first time that the AML genes are mentioned: their features and clinical associations should be better explained.
Thank you for the feed-back. We modified the section.
Lines 69-71: this section has not a conclusion.
Thank you for the feed-back. We modified the text such that it represents the aim of the study.
Lines 90-92: Results description is unclear. What are the mutations found in in each patient? Are these mutations the same? Are these mutations already known?
Thank you for the feed-back. The mutations discussed in that section refer to the TET2/IDH1/2 and WT1 mutations. TET2 and WT1 generally present loss-of-function mutations in different portions of their sequence, while IDH1/2 have few mutations described that generate the oncometabolite 2-hydroxy-glutarate. An overview of the mutations is offered by Figure 1.
Lines 103-104: delete this sentence, since it has been already stated.
Thank you for the feed-back. We removed the sentence you indicated.
Tables 2 and 3 “AML without maturation” should be listed before “AML with minimal maturation”
Thank you for the feed-back. The order follows the order in which WHO NOS are equivalent to the FAB classification of AML. Thus, we would prefer to keep the order that is currently presented.
Figure legends should be improved to be more self-explicative.
Thank you for the feed-back. We inspected the figures and the legends appear self-explicative to us considering the context of the article. If you have any specific objections we would gladly comply.
Discussion should be restructured to answer to previously raised issue. More perspectives on clinical management and therapies should be also discussed.
Thank you for the feed-back. We modified the discussions so that more emphasis was put on clinical and therapeutic management.
Reviewer 2 Report
The text deals with a topic of great interest. The classification of AML is very complex and the genetics underlying the new classification often does not translate into different therapeutic attitudes. Over 80% of patients diagnosed with AML have an intermediate cytogenetic risk and diagnostic and prognostic classification is difficult. The data reported are very interesting because they demonstrate that within the intermediate cytogenetic group, it is possible to starify patients based on hyper or hypomethylating mutations, such as IDH-1. Furthermore, the study of DNA methylation could offer patients new therapeutic possibilities,
The English language is fluent and the tables are consistent with the text.
Issue:
- from a clinical point of view it would be interesting to know how these hyper or hypomethylating mutations are associated within the classic mutations that often associate with intermediate cytogenetic risk such as CBFA, AML / ETO1, inv16, etc.
Author Response
The text deals with a topic of great interest. The classification of AML is very complex and the genetics underlying the new classification often does not translate into different therapeutic attitudes. Over 80% of patients diagnosed with AML have an intermediate cytogenetic risk and diagnostic and prognostic classification is difficult. The data reported are very interesting because they demonstrate that within the intermediate cytogenetic group, it is possible to starify patients based on hyper or hypomethylating mutations, such as IDH-1. Furthermore, the study of DNA methylation could offer patients new therapeutic possibilities,
The English language is fluent and the tables are consistent with the text.
Issue:
- from a clinical point of view it would be interesting to know how these hyper or hypomethylating mutations are associated within the classic mutations that often associate with intermediate cytogenetic risk such as CBFA, AML / ETO1, inv16, etc.
Thank you for the feed-back. The mentioned genomic alterations are underrepresented in the included cohort as most patients having these mutations are associated with intermediate cytogenetics and the mentioned mutations are part of favorable cytogenetics.
Reviewer 3 Report
The manuscript “The influence of methylating mutations on acute myeloid leukemia. Preliminary analysis on 57 patients” was aimed to hypothesize that AML patients with mutations that induce a general hypermethylation are different from those with mutations that induce a general hypomethylation .
Some comments in the paper are of interest and methods used may be innovative. However some questions persist.
The authors suggest that RNA expression are commonly only used within the research setting, but a different opinion should also be considered. Indeed, strong evidence suggest that relevant molecular mechanisms are linked to enzyme expression of gene involved in the methylation and demethylation. In this regard other observations may be associated with measurement of products of methylated or demethylated genes.
The methylation profiles, leading to alteration of protein profile often appear dependent on CpG number, a comment should be appropriate.
The authors hypothesized that AML patients with mutations that induce a general hypermethylation are different from those with mutations that induce a general hypomethylation but are similar within their own group. On the light of preliminary results which involved the effect of therapy mainly the demethylating effects, a comparison to control groups should be also added.
The tendency of CpG sites to be methylated may be regulated throughout the genome. The Authors should be able to added a comment in this issue.
Minor comments
The aim of the work at end of introduction is very similar to that reported in the abstract.
The abbreviation “OS” in the abstract should be avoided.
Author Response
Comments and Suggestions for Authors
The manuscript “The influence of methylating mutations on acute myeloid leukemia. Preliminary analysis on 57 patients” was aimed to hypothesize that AML patients with mutations that induce a general hypermethylation are different from those with mutations that induce a general hypomethylation .
Some comments in the paper are of interest and methods used may be innovative. However some questions persist.
Thank you for your appreciations.
The authors suggest that RNA expression are commonly only used within the research setting, but a different opinion should also be considered. Indeed, strong evidence suggest that relevant molecular mechanisms are linked to enzyme expression of gene involved in the methylation and demethylation. In this regard other observations may be associated with measurement of products of methylated or demethylated genes.
The methylation profiles, leading to alteration of protein profile often appear dependent on CpG number, a comment should be appropriate.
Thank you for the feed-back. We added text in the Discussions section.
The authors hypothesized that AML patients with mutations that induce a general hypermethylation are different from those with mutations that induce a general hypomethylation but are similar within their own group. On the light of preliminary results which involved the effect of therapy mainly the demethylating effects, a comparison to control groups should be also added.
Thank you for your feed-back. The aim of this study was to compare AML between themselves and not with a normal CD34+ HSC population.
The tendency of CpG sites to be methylated may be regulated throughout the genome. The Authors should be able to added a comment in this issue.
Thank you for the feed-back. We added text in the Discussions section.
Minor comments
The aim of the work at end of introduction is very similar to that reported in the abstract.
Thank you for the feed-back. We modified the text for the aim of the study.
The abbreviation “OS” in the abstract should be avoided.
Thank you for the feed-back. We changed “OS” for overall survival in the abstract.
Round 2
Reviewer 1 Report
The authors have extensively revised the manuscript according to the previous comments. The manuscript has been greatly improved.
Reviewer 3 Report
The authors have adequately responded to my suggested revisions. I find the manuscript to be acceptable.